# Predicting the functional impact of KCNQ1 variants with artificial neural networks

**Saksham Phul**[1,2], **Georg Kuenze**[1,2,3], **Carlos G. Vanoye**[4], **Charles R. Sanders**[1,5], **Alfred L. George, Jr.**[4], **Jens Meiler**[1,2,3,6]*

**1** Center for Structural Biology, Vanderbilt University, Nashville, Tennessee, United States of America,
**2** Department of Chemistry, Vanderbilt University, Nashville, Tennessee, United States of America,
**3** Institute for Drug Discovery, Leipzig University, Leipzig, Germany, **4** Department of Pharmacology,
Northwestern University Feinberg School of Medicine, Chicago, Illinois, United States of America,
**5** Department of Biochemistry, Vanderbilt University, Nashville, Tennessee, United States of America,
**6** Department of Pharmacology, Vanderbilt University, Nashville, Tennessee, United States of America

* jens@meilerlab.org

**Data Availability Statement:** Training data is available in S1 Data. All additional materials are available at https://github.com/sakshamphul/

## Abstract

Recent advances in experimental and computational protein structure determination have provided access to high-quality structures for most human proteins and mutants thereof. However, linking changes in structure in protein mutants to functional impact remains an active area of method development. If successful, such methods can ultimately assist physicians in taking appropriate treatment decisions. This work presents three artificial neural network (ANN)-based predictive models that classify four key functional parameters of KCNQ1 variants as normal or dysfunctional using PSSM-based evolutionary and/or biophysical descriptors. Recent advances in predicting protein structure and variant properties with artificial intelligence (AI) rely heavily on the availability of evolutionary features and thus fail to directly assess the biophysical underpinnings of a change in structure and/or function. The central goal of this work was to develop an ANN model based on structure and physiochemical properties of KCNQ1 potassium channels that performs comparably or better than algorithms using only on PSSM-based evolutionary features. These biophysical features highlight the structure-function relationships that govern protein stability, function, and regulation. The input sensitivity algorithm incorporates the roles of hydrophobicity, polarizability, and functional densities on key functional parameters of the KCNQ1 channel. Inclusion of the biophysical features outperforms exclusive use of PSSM-based evolutionary features in predicting activation voltage dependence and deactivation time. As AI is increasingly applied to problems in biology, biophysical understanding will be critical with respect to 'explainable AI', i.e., understanding the relation of sequence, structure, and function of proteins. Our model is available at www.kcnq1predict.org.

## Author summary

Heartbeat is maintained by electrical impulses generated by ion-conducting channel proteins in the heart such as the KCNQ1 potassium channel. Pathogenic variants in KCNQ1

KCNQ1_ML_Model. Our best performing model is available at www.kcnq1predict.org.

**Funding:** ALG, CS, JM received National Institutes of Health Research Project Grant (https://grants.nih.gov/grants/funding/r01.htm) under the grant number NIH R01 HL122010, NIH R01 GM080403. Additionally, this work in the meiler laboratory received by JM was also supported by National Institutes of Health S10 Instrumentation Program under the grant number: NIH S10 OD016216, NIH S10 OD020154 (https://orip.nih.gov/construction-and-instruments/s10-instrumentation-programs) and National Institutes of Health Research Project Grant under the grant number: NIH R01 DA046138, NIH R01 GM129261(https://grants.nih.gov/grants/funding/r01.htm). The funders had no role in study design, data collection and analysis, decision to publish, or preparation of the manuscript.

**Competing interests:** The authors have declared that no competing interests exist.

can lead to channel loss-of-function and predisposition to fatal life-threatening irregularities of heart rhythm (arrhythmia). Machine learning methods that can predict the outcome of a mutation on KCNQ1 structure and function would be of great value in helping to assess the risk of a heart rhythm disorder. Recently, machine learning has made great progress in predicting the structures of proteins from their sequences. However, there are limited studies that link the effect of a mutation and change in protein structure with its function. This work presents the development of neural network models designed to predict mutation-induced changes in KCNQ1 functional parameters such as peak current density and voltage dependence of activation. We compare the predictive ability of features extracted from sequence, structure, and physicochemical properties of KCNQ1. Moreover, input sensitivity analysis connects biophysical features with specific functional parameters that provides insight into underlying molecular mechanisms for KCNQ1 channels. The best performing neural network model is publicly available as a webserver, called Q1VarPredBio, that delivers predictions about the functional phenotype of KCNQ1 variants.

## Introduction

Congenital long QT syndrome (LQTS) is a genetic disorder of heart rhythm caused by mutations in cardiac ion channel genes [1,2] that predisposes to potentially life-threatening cardiac arrhythmia. It is among the most common genetic disorder, afflicting 1:2500 people [3]. The most prevalent subtype, LQT1, is associated with genetic variants in the *KCNQ1* gene [4,5] that encodes the pore forming subunit of the voltage-gated $K^+$ channel $K_V7.1$ (referred to as KCNQ1) [6]. In the heart, KCNQ1 forms a channel complex with KCNE1 to generate the slow delayed rectifier current, $I_{Ks}$, which is an essential driver of myocardial repolarization during the cardiac action potential [7,8]. Pathogenic variants that cause KCNQ1 loss-of-function (LOF) lead to diminished $I_{Ks}$ and impaired repolarization that is manifest by prolongation of the QT interval on surface electrocardiograms [9].

LQT1 is among the most common inherited disorders [10]. More than 1000 genetic variants of KCNQ1 have been identified [11,12], but for many variants there are insufficient data to classify each as either pathogenic or benign. Correlating these variants of uncertain significance (VUS) to their clinical outcomes and determining the risk of LQTS remain major challenges [13,14]. Large-scale functional characterization of KCNQ1 variants has been made feasible by using automatic patch-clamp recording [15] and this strategy helped reclassify variants with previously conflicting or unknown interpretations according to the ClinVar database [11]. Moreover, the mechanistic basis underlying mutation-induced KCNQ1 dysfunction has been investigated [16–20]. For instance, Huang et al. [19] studied the impact of mutations in the KCNQ1 voltage-sensing domain (VSD) on protein cell surface expression, trafficking, protein folding, and structure. More than half of LOF mutations examined were found to destabilize the VSD structure resulting in impaired trafficking and lower cell surface expression. This observation underscores the growing notion that mutation-induced destabilization and mistrafficking of the KCNQ1 protein are common disease mechanisms in LQT1. However, this study also identified LOF variants that did not exhibit trafficking and folding defects, indicating heterogeneity in the molecular mechanisms responsible for KCNQ1 LOF that cause LQT1. The molecular function of many variants in other regions of the KCNQ1 channel have yet to be characterized, and it is expected that these investigations will reveal additional pathogenic mechanisms [21].

Despite this progress in functional characterization of KCNQ1 variants, experimental assays remain labor-intensive, and this limits their applicability in a clinical setting. Computational approaches can support experimental testing and have the potential to help elucidate the molecular function of KCNQ1 variants as well as predicting associated clinical outcomes [22–25]. Computational methods trained on information from genome-wide genetic variation data are commonly used for protein variant effect prediction [26–30], but these tools have limited applicability for KCNQ1 (see Table S4 in reference [23]). Prediction accuracy of genome-wide methods is low and varies between targets. This reflects that development of these methods was based on heterogenous datasets including a wide range of proteins with diverse functions and associated diseases. Furthermore, these methods fail to establish the precise effect of a variant on KCNQ1 function parameters [23]. To overcome these difficulties, specific machine learning models tailored to predict the functional effects of KCNQ1 variants were developed [23,24]. Similar approaches have also been applied to other cardiac ion channels [31,32].

KCNQ1 is most often associated with autosomal dominant LQTS, and rarely with recessive LQTS. Dominant-negative loss-of-function (LOF) mechanisms have been implicated in autosomal dominant LQT1. Patients with heterozygous mutations are associated with autosomal dominant LQTS (Romano–Ward syndrome) whereas patients with homozygous or compound heterozygous KCNQ1 mutations have a more severe clinical outcome and are associated with recessive LQTS (Jervell–Lange–Nielsen syndrome)[33]. Even for the common heterozygous forms of LQTS, it is valuable to predict the function of a variant in the homozygous state; mainly for discriminating benign from pathogenic variants. This knowledge contributes to determining the disease-causing propensity of variants found in recessive LQTS and identifies variants with potential to cause autosomal dominant LQTS.

Q1VarPred [23] is a KCNQ1-specific channel function predictor. Criteria for dysfunction were calibrated by examining experimentally determined electrophysiology parameters for KCNQ1 (i.e., peak current density, voltage of half-maximal activation) that were then used to train a neural network with input features derived from protein sequence. Q1VarPred achieved greater accuracy than genome-wide tools, which perform poorly for membrane proteins [34]. Additional predictive power may be gained by analyzing the spatial clustering of variants in the 3-dimensional structure of KCNQ1 [31]. Functionally critical channel regions, such as the ion selectivity filter and cytosolic gate in the pore domain (PD) and the S4 helix in the VSD, are "hotspots" for variants causing the greatest perturbations in peak current density and voltage of activation, respectively. This suggests that protein structure features can aid variant prediction.

In this study, we used machine learning to develop an KCNQ1 variant prediction tool called Q1VarPredBio (www.kcnq1predict.org). The functional classification categories of Q1VarPred were expanded by a scheme that predicts variant-specific changes in four electrophysiological KCNQ1 parameters: peak current density, voltage of half-maximal activation ($V_{1/2}$), and activation and deactivation time constants ($\tau_{act}$, $\tau_{deact}$). We evaluated the performance of artificial neural networks (ANNs) trained on evolutionary and biophysical features for KCNQ1 and observed that a combination of both features produced a model with optimal predictive accuracy. Our machine learning approach can be useful to obtain insights into basic sequence-structure-function relationships for the KCNQ1 channel. Moreover, Q1VarPredBio may help differentiate between potential pathogenic dysfunctional KCNQ1 variants from those with normal channel function.

## Results

We developed three types of ANN models: one trained with only evolutionary features, one trained with biophysical features, and a third one with both evolutionary and biophysical

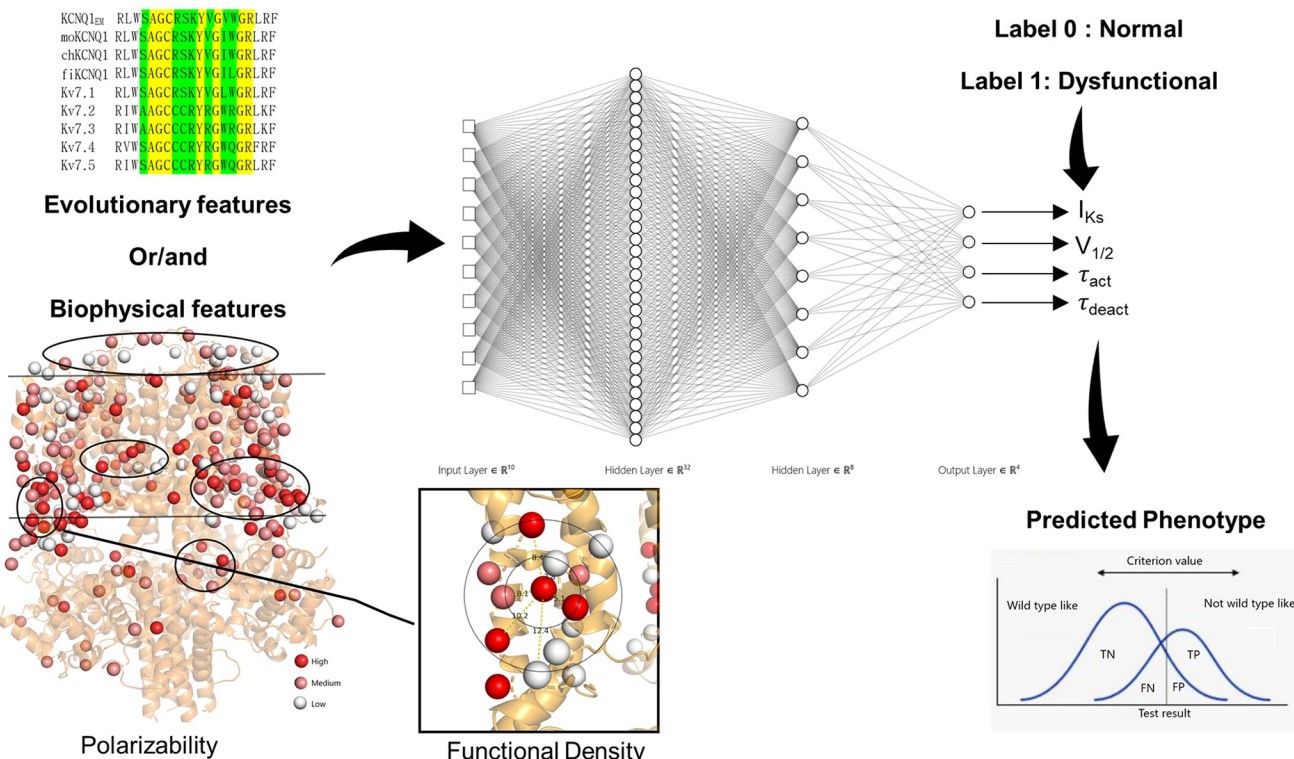

**Fig 1. A Schematic flowchart of the ANN Models.**

features. All three models were trained to predict the mutation-induced change in four functional parameters (peak current density, voltage of half-maximal activation, activation, and deactivation time constants) of the KCNQ1 channel as either normal (label 0) or dysfunctional (label 1) phenotype. All ANN models had an input layer, two hidden layers, and an output layer with four neurons. A schematic representation of our model development workflow is shown in Fig 1. These multitask ANN models were trained to improve accuracy of predicting all parameters combined. All models predicted four outputs between 0 and 1. A decision boundary was identified at the best possible accuracy and presented our threshold to classify a variant parameter as being either normal or dysfunctional. Accurate prediction based on this decision boundary was a criterion for determining the model performance. To measure the performance of the model, we adopted a 25-fold cross validation technique wherein performance was evaluated on a variant test set that was not included in model training and monitoring. Further details on ANN architecture, training, and model performance are described in the methods and materials section.

## Dataset and criteria

The dataset for this work contained electrophysiological data for 125 KCNQ1 variants that were generated, tested, and functionally analyzed using the same approach [15,35]. The KCNQ1 variants were tested in the homozygous state transiently co-expressed with wild type KCNE1 in CHO cells. The electrophysiological data measured for each variant consists of four biophysical parameters: peak current density (designated as $I_{Ks}$), voltage of half-maximal channel activation ($V_{1/2}$), activation time constant ($\tau_{act}$), and deactivation time constant ($\tau_{deact}$). In

**Table 1. Criteria for classifying variants as dysfunctional used in this work.**

| Loss of function (%WT) | Phenotype | Gain of function (%WT) |
|---|---|---|
| < 55% | $I_{ks}$ | > 115% |
| > 130% | $V_{1/2}$ | < 80% |
| > 170% | $\tau_{act}$ | < 70% |
| < 75% | $\tau_{deact}$ | > 125% |

order to compare the functional properties of variants tested across many months, the values for each parameter were normalized to the values obtained from cells expressing the wild type channel that were transfected and tested in parallel. Normalized values equal to 1 (or 100% WT) were considered wild-type-like. A parameter phenotype was classified as dysfunctional if it satisfies the criteria in Table 1. These thresholds were derived collectively from values defined in Li et al [23], disease-causing variants in literature and by evaluating model performance at different thresholds.

For training and testing of our model, both gain-of-function and loss-of-function (according to Table 1) were classified as dysfunctional. Biophysical parameters that could not be determined for some variants (e.g., voltage-insensitive [no $V_{1/2}$] variants that do not deactivate [no $\tau_{deact}$]), were defined as dysfunctional and for variants with peak current density $\leq$ 17% WT, all four biophysical parameters were considered dysfunctional.

Due to the scarcity of the functional data from certain protein regions, we introduced 345 'non-perturbing' variants, one for each of the 345 amino acids included in the KCNQ1 structural model [36] thereby increasing the size of our dataset to 470 (see S1 Data). All four functional parameters for the non-perturbing variants were considered WT. These non-perturbing variants expose ANNs to all the structural regions of the protein during training. This helps the model to recognize changes in structure and physicochemical properties at the site of mutation for all neighborhoods that exist in the structure. The extent of these changes helps ANNs to classify the phenotype of a mutation (i.e., benign or pathogenic). These non-perturbing variants create a 'baseline' for the protein region where data was limited allowing the ANNs to train on a greater number of instances. In summary, there were 345 non-perturbing, 39 benign and 86 pathogenic variants based on the peak current criteria given in Table 1.

## Identification of biophysical features

In total, we used 14 structural and physicochemical properties of KCNQ1 to develop an ANN model based solely on biophysical features. These features were extracted by importing the KCNQ1 structure into Biochemical Library (BCL) software. The KCNQ1 cryo-EM structure model utilized for this work, had bound calmodulin (CAM), no $PIP_2$ and represented a decoupled state with activated voltage sensor domain and a closed pore domain [36]. We explored different biophysical features based on existing understanding of molecular mechanisms underlying KCNQ1 function and the location of critically significant regions in the KCNQ1 structure important for protein stability and channel gating. These biophysical features inform about the amino acid local environment, exposure to solvent, burial in the membrane, change in amino acid physicochemical properties at the mutation site, steric hindrances near to the α-carbon atom, and the mutation-induced change in water-membrane transfer free energy. Distance from the KCNQ1 channel pore axis was used to help the model distinguish variants in the channel pore domain from those in the voltage sensor domain (see S1 Fig). Furthermore, it

is more likely that a variant residue buried in the membrane will negatively impact protein function. Thus, the degree of burial of a variant in the membrane was assessed by using a three-layered membrane model and calculating a membrane-depth dependent weight calculated with a distribution function (S1 Eq 1). A significant weight was given to variants embedded inside the membrane (see S2 Fig). Steric hindrances near the α-carbon atom were examined using a steric parameter (see S1 Table), which is a graph shape index that encodes complexity, branching and symmetry of amino acid side chain [37].

Transfer free energy of an amino acid between a hydrophilic and hydrophobic environment plays a crucial role for protein folding and stability. Thus, we used hydrophobicity of native and variant amino acids to investigate the mutation-induced changes in the free energy for transfer into the membrane (see S3 Fig). The mutation-induced change in water-membrane transfer free energy at an amino acid site was examined using the hydrophobicity scale reported by Koehler *et al* [38]. Highly hydrophobic or hydrophilic amino acids are usually surrounded by similarly hydrophobic or hydrophilic amino acids. Thus, we calculated the polarizability and hydrophobicity of amino acids at and around the variant site by functional density [31]. Functional density is based on k-nearest neighbors' algorithm, wherein the average physiochemical property around the site of variant is weighted by the inverse of their distance from the site of mutation (see S4 Fig). These features examined hydrophobicity and polarizability of the neighborhood around the site of variants. We found that there exists a correlation between high polarizability regions in the protein and dysfunctional peak current density (see S5 Fig). More details on the implementation of these biophysical features are described in the S1 Text.

We also introduced changes in the physicochemical properties [number of hydrogen bond donor sites, number of hydrogen bond acceptor sites, and van der Waals volume [37]] of the amino acid at the site of mutation, in order to help the ANN model learn whether an amino acid substitution represents a missense or non-perturbing mutation. These properties also improved the functional outcome predictions for missense mutations.

Exposure of variant sites to solvent was quantified using the neighbor vector method [39] (see S6 Fig). Neighbor vector [39] improved the predictions especially for peak current density whereas neighbor count [39] did not. Backbone conformation (Phi (φ), Psi (ψ), and Omega (ω) angles) for the native amino acid and other descriptors like the location of a mutation on a helix, mutation-caused change in amino acid polarizability as well as change in hydrophobicity and solvent accessible surface area for amino acids did not improve prediction accuracy.

### Evolutionary feature: PSSM-based amino acid substitution score

We used PSI-BLAST search to calculate a position-specific scoring matrix (PSSM) which measures the likelihood of amino acid substitution at a mutation site [40]. PSSM was created by searching UniRef50 [41] and the NCBI non-redundant sequence databases [42]. The difference of PSSM scores between variant and WT amino acid from these two databases was utilized as evolutionary features. We found that these evolutionary features were solely sufficient in predicting the functional properties of non-perturbing mutations. More details can be found in the S1 Text.

### Biophysical features outperform PSSM-based evolutionary features in predicting activation $V_{1/2}$ and $\tau_{deact}$

Model accuracy was evaluated using Matthew's correlation coefficient (MCC) and receiver operating characteristic (ROC) plots by testing a variant set that was omitted from model training and monitoring. A decision boundary was identified at the best possible accuracy

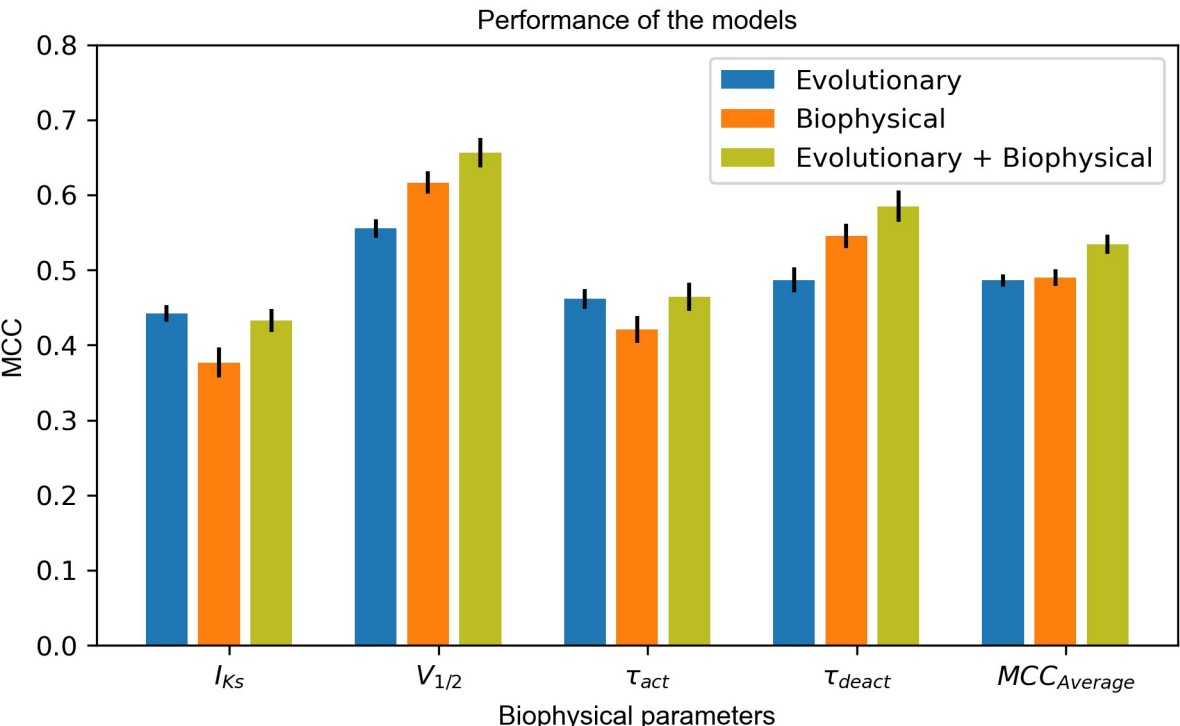

**Fig 2. MCC values reported for three different ANN models.** The error bar represents 1x standard deviation.

(MCC value) whereas ROC plots were independent of decision boundary. These MCC values and ROC plots for different ANN models are reported in Figs 2 and 3 respectively. More details on MCC and ROC are described in Methods and Material section.

We were able to model $V_{1/2}$ and $\tau_{deact}$ using PSSM-based evolutionary features, biophysical features, and both features combined. MCCs for these feature sets are reported in Fig 2. Although evolutionary features achieved satisfactory performance in predicting $V_{1/2}$ with a MCC of 0.56, biophysical features perform better by attaining MCCs greater than 0.62. The area under the curve (AUC) for $V_{1/2}$ determined for biophysical features was greater than that for PSSM-based evolutionary features (Fig 3). For $\tau_{deact}$, evolutionary features achieved satisfactory performance in predicting $\tau_{deact}$ with a MCC of 0.50, biophysical features perform better by attaining MCCs of greater than 0.56. Area under the curve (AUC) for $\tau_{deact}$ determined for biophysical features was greater than that for evolutionary features (Fig 3). Biophysical features clearly dominate in predicting the activation $V_{1/2}$ and $\tau_{deact}$ suggesting that the ANN model could determine and distinguish structure-activity relationships underlying the voltage-dependence of KCNQ1 activation and deactivation of KCNQ1 kinetics.

## Biophysical features can predict current density but do not outperform PSSM-based evolutionary features

We were able to model peak current density ($I_{Ks}$) using evolutionary and biophysical features. For peak current density ($I_{Ks}$), biophysical features attained MCC close to 0.38 whereas PSSM-based evolutionary features achieved MCC $\geq$ 0.43, suggesting that PSSM-based amino acid substitution scores perform better in predicting peak current density. Moreover, the AUC for

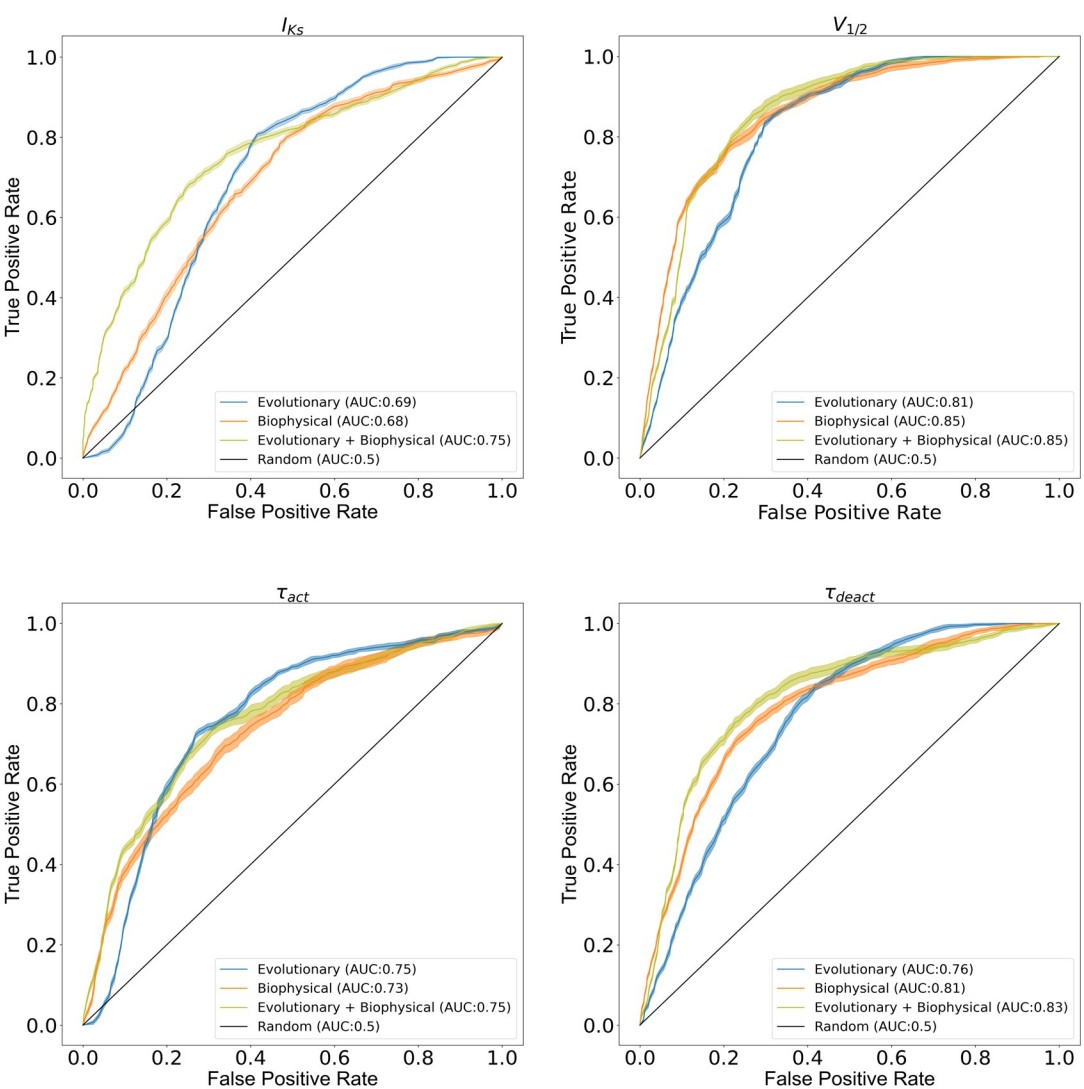

**Fig 3. ROC Plots with 99% confidence interval (shaped region) for all-functional parameters of KCNQ1 channel considered in this study.** Different color scheme is used to depict performance of the three feature sets.

biophysical features is 0.68 whereas for evolutionary features is 0.69 (see Fig 3). This suggests that biophysical features are comparable with PSSM-based evolutionary features in distinguishing between normal and dysfunctional variants. BLAST based PSSM-derived amino acid substitution scores, which is our evolutionary feature, has a significant association with peak current density ($I_{Ks}$) as previously reported by Kroncke *et al* [31].

## PSSM-based evolutionary features predict $\tau_{act}$ better than biophysical features

We modeled $\tau_{act}$ using both biophysical and evolutionary features. Evolutionary features yielded a MCC of 0.44 whereas biophysical features yielded a MCC of 0.40, suggesting evolutionary features perform better than biophysical features. Similarly, AUC for evolutionary features was close to 0.75 better than that for biophysical features (AUC:0.73). We observed that

the ANN model makes better predictions for peak current when activation time labels are simultaneously present in the training dataset.

## Performance of biophysical features is comparable with PSSM-based evolutionary features

To compare models using a common performance metric, the MCCs calculated for four functional parameters were averaged ($MCC_{average}$). The $MCC_{average}$ for biophysical features was 0.49 wheras $MCC_{average}$ for evolutionary features was 0.48 (Fig 2). Thus, the two models are comparable in combined performance, irrespective of their performance for the individual functional parameters. The average AUC for biophysical features was 0.77 whereas evolutionary features was 0.75, suggesting biophysical features are comparable with PSSM-based evolutionary features.

## Most accurate predictions are achieved by combining biophysical and PSSM-based evolutionary features

For training the ANN model with both biophysical and evolutionary features, we used eleven biophysical features and the difference of PSSM score determined with NCBI non-redundant sequence database ($\Delta PSSM(NR)$). We found that three [Neighbor Vector, Mutant steric parameter, and Native steric parameter] out of 14 features used for the biophysical model and uniref50 based PSSM scores failed to improve the predictions on the unseen dataset, therefore these features were excluded (see S9 Fig). This could be due to redundancy in the information carried by these three biophysical and evolutionary features. The eleven biophysical features included were: hydrophobicity of mutant amino acid, polarizability of mutant amino acid, functional density of amino acid polarizability with neighborhood sizes of 6.5 Å and 12 Å, functional density of amino acid hydrophobicity with neighborhood sizes of 1 Å and 6.5 Å, change in number of hydrogen donor sites, change in number of hydrogen acceptor sites, change in Van der Waals volume, distance from the pore axis, and depth of the mutation site in the membrane.

MCC > 0.44 and an AUC > 0.75 for peak current density suggest that although evolutionary and biophysical features combined do not improve prediction accuracy (MCC) for peak current density they do improve the spread (AUC) between normal and dysfunctional variants predicted by the model. Moreover, significant improvement in performance was observed for $V_{1/2}$ and $\tau_{deact}$ by combining evolutionary and biophysical features. MCC increases by 15% for $V_{1/2}$ and by 10% for $\tau_{deact}$ when compared with the single-feature best model, i.e., biophysical model. The results also suggest that biophysical features have a significant association with $\tau_{deact}$ and $V_{1/2}$ making the performance of biophysical features for these parameters better than that of evolutionary features. For $\tau_{act}$, we observed no improvement in MCC or AUC (Figs 2 and 3) for the combined model when compared with evolutionary features, indicating the redundant nature of evolutionary and biophysical features in capturing sequence-structure-function relationships about KCNQ1 $\tau_{act}$. Overall, in the combined features ANN model, $MCC_{average}$ increases to 0.54, which corresponds to a 12% improvement in $MCC_{average}$ relative to the best single-feature model.

## Wildtype like variants were slow to activate

We observed that variants H105N, T118S, V129I, and E146G had peak current density, $V_{1/2}$, and $\tau_{deact}$ parameters close to WT values but significantly larger activation time constants. Additionally, it was difficult to model $\tau_{act}$ with any of the ANN models using a threshold 80%–

120% of normal. Based on those observations, we adjusted our threshold for normal $\tau_{act}$ to 70%–170% of wild type. This change improves MCC for PSSM-based evolutionary features from 0.22 to 0.44 and yielded 10% improvement in average MCCs across all models.

## Performance of ANN models on non-perturbing variants

The MCC and AUC reported in Figs 2 and 3 were evaluated on 125 experimentally validated variants. The correct classification of non-perturbing mutations signifies that ANNs can distinguish no change in protein structure results into a benign variant. This is an important feature as the wild-type sequence is not directly input into the ANN but encoded as changes in evolutionary and biophysical parameters. Thus, these non-perturbing mutations ensure that a change of zero in all parameters are understood as wild-type. All models were able to classify roughly 98% of the 345 non-perturbing variants accurately as benign. Features like change in number of hydrogen donor sites, change in number of hydrogen acceptor sites, and change in amino acid volume were required for an ANN model trained with only biophysical features to help model learn non-perturbing mutations. When both features were combined, evolutionary features were found to be sufficient in correctly predicting non-perturbing mutations. The accuracy of the combined ANN model for non-perturbing, benign, and pathogenic variants is shown in S7 Fig. This figure depicts that ANN can recognize these three types of variants and predict majority of them at different regions between 0 and 1. Additionally, to quantify the separation between non-perturbing and benign mutations, we calculated entropy at 0.05 as the decision boundary to separate non-perturbing from benign variants. The lower the entropy (best~0, worst~1), the better is the separation of the two classes. Entropy for all the functional parameters was less than 0.35 suggesting the model can distinguish between non-perturbing and benign variants.

## Predictive ability based on the function of the variants

Using peak current density to define variant classes, there were 71 LOF, 15 GOF and 39 WT-like variants in the functional dataset. All ANN models predict LOF better than GOF or WT-like (see S8 Fig). Biophysical features-based model predicts GOF better than other feature sets whereas evolutionary-based model predicts LOF better than the rest. By combining biophysical and evolutionary features, there is a significant improvement in predictive ability of WT-like variants.

## Input sensitivity highlights feature importance and their association with KCNQ1 functional parameters

To study the contribution of different features in our ANN model, we examined the input sensitivity of input features on output labels. Since considering the magnitude of input sensitivity for feature importance can be misleading due to the issue in rescaling the input features [43], we considered sign of the input sensitivity with respect to output label. Input sensitivity is defined as zero when half instance of the variants predicts a positive change with respect to the result label and the other half predict a negative change with respect to the result label. Similarly, an input sensitivity close to one signifies that the input feature strongly correlates with the output label. More details on input sensitivity analysis can be found in the Materials and Methods section.

The input sensitivities for our best predictive model (eleven biophysical features and one evolutionary feature) are averaged across 320 models with each model having 25 instances of input sensitivity using 25 different monitoring data subsets. These averaged input sensitivities are reported in Fig 4.

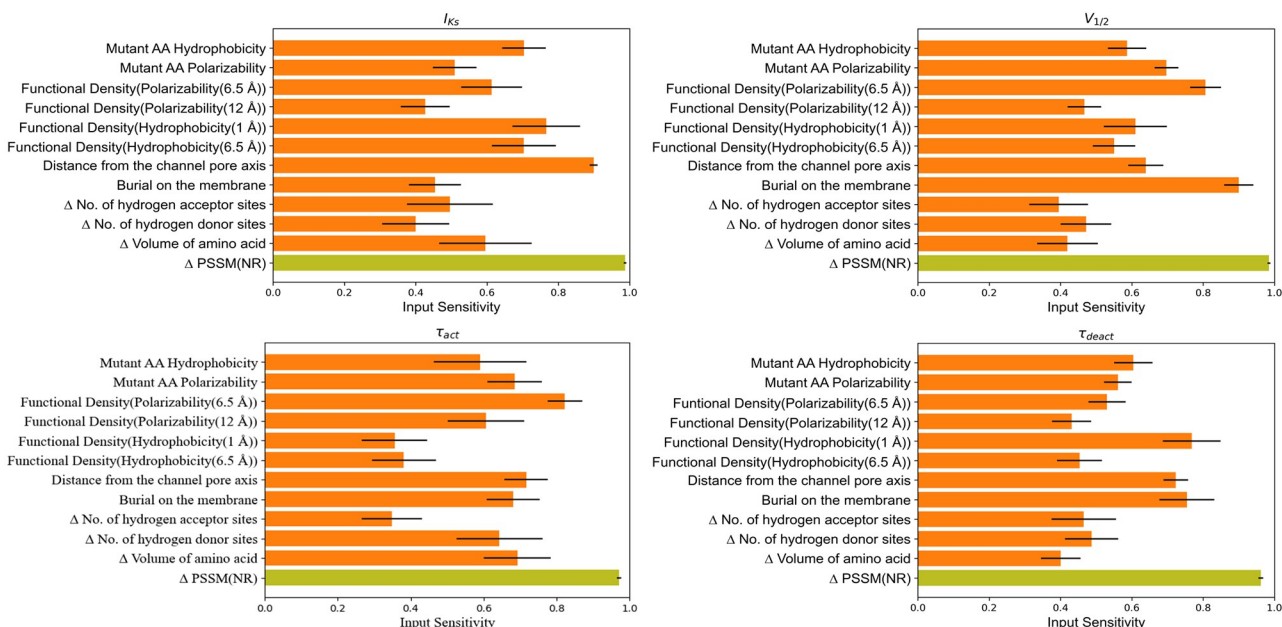

**Fig 4. Average input sensitivities for ANN model trained with combined evolutionary and biophysical features.** Biophysical features are shown in orange and evolutionary features in olive color.

PSSM-based amino acid substitution score ($\Delta$ PSSM(NR)) was found to be the most sensitive feature for all the functional parameters, suggesting that sequence-based features are of high quality. For peak current density ($I_{Ks}$), four biophysical features were highly sensitive: hydrophobicity of mutant amino acid, hydrophobicity around mutation site within radius of 1 Å and 6.5 Å, and distance of mutation site from channel pore axis. For predicting $V_{1/2}$, the most sensitive biophysical features were burial of mutation site in the membrane, mutant polarizability, and neighborhood polarizability around mutation site with 6.5 Å radius. For $\tau_{act}$, the most sensitive biophysical properties were neighborhood polarizability around mutation site with 6.5 Å radius, distance from the channel axis and change in volume of the amino acid. And for predicting $\tau_{deact}$, the most sensitive features were burial of mutation site on the membrane, distance of mutation site from channel pore axis, and neighborhood hydrophobicity around mutation site within 1 Å radius.

We also report the input sensitivity for an ANN model trained solely with 14 biophysical features. These input sensitivities are also averaged across 320 models with each model having 25 instances of input sensitivity using 25 different monitoring data subsets. The averaged input sensitivities for the biophysical features only model is reported in Fig 5. We observed that all biophysical features are sensitive to functional parameters, but some features were more sensitive to specific functional parameters than other biophysical features. The distance from the channel pore axis, change in number of hydrogen donor sites and hydrophobicity at neighborhood size of 1 Å were the top highly sensitive biophysical features for $I_{Ks}$. The mutant amino acid hydrophobicity, neighbor vector and burial on the membrane for $V_{1/2}$. Change in number of hydrogen donor sites and mutant amino acid hydrophobicity for $\tau_{act}$ and burial on the membrane and neighborhood polarizability within 12 Å radius for $\tau_{deact}$.

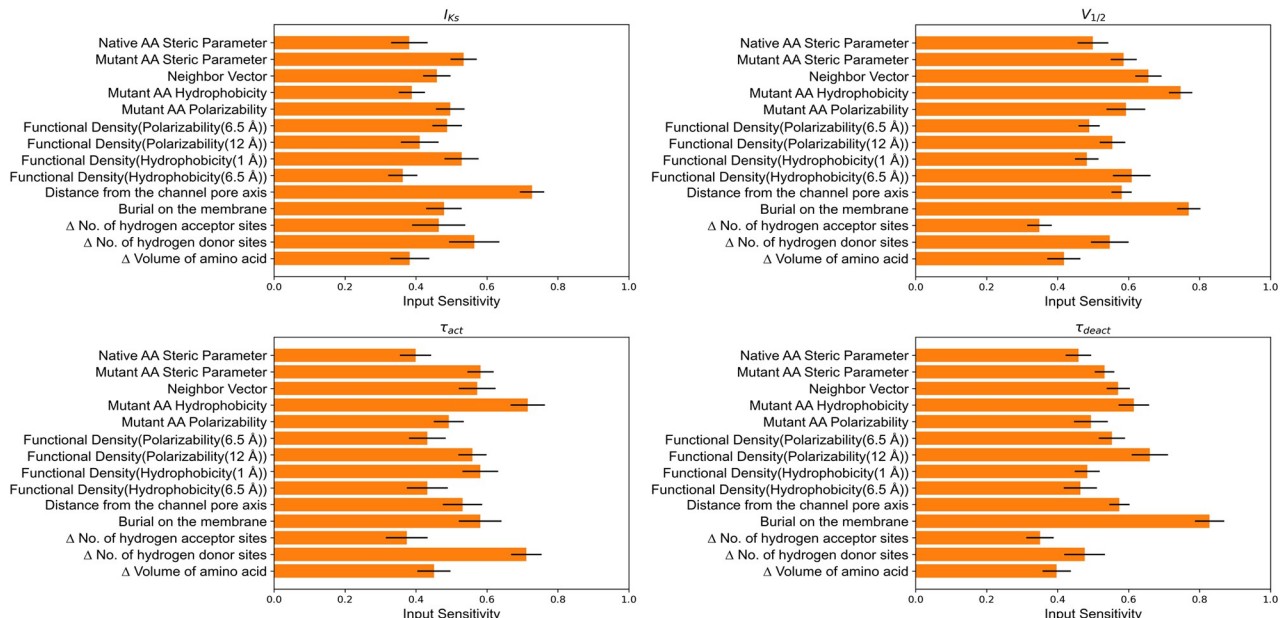

**Fig 5. Average input sensitivities for ANN model trained with only biophysical features.**

## Discussion

### A note on training the ANNs on functional data tested in the homozygous state

The paper in Science Advances 6, 2018 by Huang et al. provides evidence that training models based on experimental data collected on homozygous cells is relevant for KCNQ1 related diseases [19]. Specifically, this group conducted experiments in which WT KCNQ1 was co-expressed with a mutant of interest and the total trafficking of KCNQ1 was quantitated (see Figure 3 in reference [19]). They found that the total and cell surface expression of total KCNQ1 (WT + mutant) was usually in between the results for WT only and mutant-only. However, there were a few exceptions in which the WT protein appeared to rescue trafficking of the mutant or where the mutant protein impeded trafficking of the WT protein. These results suggest that studies of mutant-only condition are usually a good predictor for the corresponding WT/mutant heterozygous conditions, but there are exceptions.

### Biophysical features perform as well as PSSM-based evolutionary features in predicting KCNQ1 variant function

Biophysical features performed as well as evolutionary features in predicting the functional outcomes of KCNQ1 variants. This suggests that ANN with only biophysical features only recognized relationships between KCNQ1 structure, function, and mutation-induced dysfunction. However, biophysical features outperformed the PSSM-based amino acid substitution score in predicting the $V_{1/2}$ and $\tau_{deact}$. This could be linked to the prevalence of variants located in the VSD amongst all variants in the training dataset, thus, allowing the network to effectively learn about this channel domain. We found that presence of $V_{1/2}$ labels in the training dataset improves the $\tau_{deact}$ predictions and that biophysical features that were highly sensitive to $V_{1/2}$, were also found to be sensitive to $\tau_{deact}$ (Figs 4 and 5). These observations could

indicate that similar molecular determinants are important for voltage-dependent channel activation as well as kinetics of channel deactivation. Biophysical features like burial on the membrane and distance from the channel pore axis were most important for the ANN to learn the phenotype of variants in the VSD. Other features like the polarizability of mutant amino acid and the functional density of amino acid polarizability within a radius of 6.5Å and 12Å around the mutation site were also sensitive to $V_{1/2}$ and $\tau_{act}$ (see Fig 4). This could indicate the fact that high polarizability of amino acids in the VSD confers sensitivity to transmembrane voltage that is required for KCNQ1 activation mechanisms. For instance, at R195 the functional density of polarizability is 0.19 for radius size of 6.5 Å. Mutation at this site to Q and P decreases site polarizability. R195Q and R195P exhibit LOF possibly because changed amino acid polarizability affects KCNQ1 activation. This is in line with the high sensitivity of $V_{1/2}$ prediction for polarizability as shown in Fig 5. Similarly, R195W is a GOF mutant and this correlates with increase in the site polarizability. However, not all the sites in VSD actively participate in activation; some exist to stabilize the protein fold of the VSD [19]. L114P, E115G, Y125D, G189A, S225L, L236P, and L236R are variants that fail to activate due to protein misfolding [19]. These variants have similar native hydrophobicity and neighborhood hydrophobicity, however, mutations at these sites result in changes in hydrophobicity suggesting that the amino acid side chains for these variants are located in an energetically unfavorable environment. Therefore, network predictions have high sensitivity for mutant hydrophobicity, and functional density for hydrophobicity, suggesting that protein stability is a crucial aspect in interpreting the functional phenotype of $V_{1/2}$ and $\tau_{deact}$. Exposure of a site (neighbor vector) to solvent environment is also sensitive to functional phenotype of $V_{1/2}$ and $\tau_{deact}$.

We were able to model peak current density and $\tau_{act}$ using only biophysical features, however, this model performance was not as good as the one obtained with only PSSM-based evolutionary features. In summary, it was challenging to find biophysical features that significantly improve the peak current density and $\tau_{act}$ predictions over PSSM amino acid substitution scores. Interestingly, we observed some interdependency of peak current density and $\tau_{act}$ predictions, similar as for $V_{1/2}$ and $\tau_{deact}$. Due to the multi-task classification scheme, the network model can benefit and learn from other predicted labels, thus, increasing prediction accuracy compared to a single-task classification network. As per our dataset, it is more probable for $V_{1/2}$, $\tau_{act}$ and $\tau_{deact}$ to be dysfunctional if peak current is also dysfunctional. This is the case for 42 variants in the training set, for which $I_{Ks}$ is less than 17% WT giving rise to dysfunctional $V_{1/2}$, $\tau_{act}$ and $\tau_{deact}$. In summary, biophysical model performs well in predicting variant phenotype when all four parameters are dysfunctional or normal, but it does not perform well when only one or two parameters are impaired.

## Combination of PSSM-based evolutionary and biophysical features improve functional phenotype predictions

When combining both types of features, the ANN model predicts functional parameters with better average accuracy than models trained with only biophysical or evolutionary features. For the voltage-dependence of activation $V_{1/2}$, prediction accuracy markedly improves by combining both feature sets. On examination of the input sensitivities in Fig 4, functional density of polarizability within 6.5Å radius, burial on the membrane, and mutant polarizability are the biophysical features that have highest impact on $V_{1/2}$ prediction. The movement of helices in VSD under the influence of electric field is due to the sites with high polarizability residues and neighborhoods with high polarizability. Therefore, mutant polarizability and functional density of polarizability are biophysical features with high sensitivity and can be linked to the KCNQ1 activation mechanism. Burial on the membrane indicates higher prominence of

mutations that are inside the membrane and in close proximity to VSD and PD region. For example, variants R259H, V280E, A300E, A300T, F340L, A344V and others are located in membrane-embedded regions of KCNQ1 and near to the pore domain cause LOF. On the other hand, site A287 is in close proximity to the pore but lies outside the membrane. This could explain why A287E and A287V variants are WT-like for all functional parameters. Proximity to the pore and location in the membrane appear to classify sites that are in functionally critical regions of the protein.

Prediction accuracy for $\tau_{deact}$ increases when biophysical and evolutionary features are combined, MCC improves to 0.60, and AUC increases to 0.83. Based on input sensitivity (see Fig 4) burial on the membrane, distance from the channel pore, mutant hydrophobicity, and functional density of hydrophobicity within radius 1 Å were highly sensitive biophysical features for $\tau_{deact}$ prediction. The high sensitivity for hydrophobicity-based features indicates that stability at the site of the mutation affects the phenotype of $\tau_{deact}$. The improvement in MCC and AUC scores for $\tau_{deact}$ may be due to the improved ability of the model to identify the VSD region by using biophysical features like burial on the membrane and distance from the channel pore axis. We also observed that $\tau_{deact}$ predictions improved when $V_{1/2}$ labels were present, highlighting to a relation between $V_{1/2}$ and $\tau_{deact}$.

For peak current density, there was a small improvement in MCC, suggesting that for this parameter, both feature types carry similar information content. Based on our input sensitivity analysis for peak current density shown in Fig 4, we deduce that evolutionary features (i.e., PSSM score) carry valuable information about protein folding stability. Even though hydrophobicity-based features are highly sensitive features for peak current density, adding them to the network training process did not improve performance. This means that PSSM scores, and hydrophobicity feature carry redundant information for predicting peak current density. The fact that hydrophobicity features like mutant hydrophobicity and functional density with hydrophobicity are amongst the most sensitive features highlights the relationship of peak current density with protein structure stability. It is possible that due to protein structure instability, KCNQ1 protein tends to misfold resulting into dysfunction. The work from Huang et al. observed that many variants in the KCNQ1 VSD negatively impact protein folding stability, leading to trafficking defect and consequently low peak current density [19]. Considering that VSD region variants predominate in our database and the high input sensitivity for hydrophobicity-based features, we also conclude that protein stability is a major effect of mutations in VSD.

Combining biophysical and evolutionary features did not improve the MCC for $\tau_{act}$ whereas AUC increased from 0.81 to 0.83. Among the different biophysical features used for predicting $\tau_{act}$, we found that the change in the number of hydrogen donor sites due to amino acid substitution significantly improved the predictions for $\tau_{act}$. Likewise, the change in the number of hydrogen bond donor sites was also found to be sensitive to $\tau_{act}$. There were 43 variants in our dataset that experienced loss of donor sites due to amino acid substitution. Possible explanations for the number of hydrogen bond donor sites and $\tau_{act}$ association are that $\tau_{act}$ is dependent on the hydrogen bonds assisting in the activation mechanism in the VSD region of the protein, or these hydrogen bonds interact with PIP2 complex in sending a signal to the pore domain. Sun *et al.* found that residues near the S4 helix and S4-S5 linker helix interface interact with $PIP_2$[36]. $PIP_2$ has charge -3, -4 or -5 depending on pH of the surroundings, and the presence of the negative charge makes PIP2 a proton acceptor. Interestingly, the majority of the sites such as R116, R195, K196, Y184, K183, R181, and R249 are located within 4 Å distance from the $PIP_2$ interacting sites at the S2-S3 linker, S3 helix, S4 helix, and S4-S5 linker that are proton donors and have high polarizability. Thus, mutations in this protein region could lead to impairment in the function of the

KCNQ1 protein due to the loss of hydrogen donor sites, impacting regulation by $PIP_2$. The loss of hydrogen donor sites and consequently hydrogen bonds, can also be linked to stability at the mutation site. The high sensitivity of $\tau_{act}$ for mutant hydrophobicity (see Figs 4 and 5) indicate protein destabilization as a likely cause for impairment. Furthermore, the high sensitivity to change in amino acid side chain volume highlights the structural impact on local environment near the site of mutation. $\tau_{act}$ was also found to be sensitive to polarizability-based features highlighting the role that this feature places as driving force (response by amino acids under electric field) in ion channel activation.

## Significance of this study

This work demonstrates the capability of ANN models and biophysical features to predict the phenotype of four KCNQ1 functional parameters. Recently, explainable AI has been an important consideration for usage of AI in medicine [44,45]. We argue that ANNs trained on biophysical rather than PSSM-based evolutionary features enable understanding of determinants of function in a more transparent way. Moving forward with AI in biology, this understanding will be critical with respect to explainable AI, i.e., understanding the relation of sequence, structure, and function of proteins. Moreover, input sensitivity analysis link biophysical features with functional parameters providing insights on underlying molecular mechanisms.

## Limitations and future directions

The primary limitation of this study is the size of the dataset and overrepresentation of variants from the VSD. There exists a substantial amount of functional data available in literature, however, those results were generated and analyzed using different expression and testing systems which may complicate model training. Our models were trained on data obtained with the same experimental approach and measurement protocols. Although we were able to train the models with 125 variants, there is a need for more functional data especially for variants in the pore domain (PD). Another limitation of this study is that variants were tested in the homozygous state. This work is sufficient for evaluating disease-causing propensity of variants found in severe cases with recessive LQTS but might not address all cases of dominant LQTS. Moreover, the incomplete KCNQ1 structure limits the ability of our models to make predictions for only half of amino acid sites ($< 350$ sites), mostly localized to the S1-S6 region.

An interesting question is whether analyzing multiple, possible functional conformations of the KCNQ1 protein, modeled by Monte-Carlo or molecular dynamics simulations, can provide an orthogonal set of information which can then be used to further improve the ANN model. There is experimental evidence that the KCNQ1 VSD can exist in one of three states (resting, intermediate, activated), which are coupled to the pore domain and influence opening. Experimental and model structures are available for these states [46–47]. By incorporating structural and biophysical data about those states, our ANN model could learn molecular properties that underlie ion channel gating and how these properties are changed by variants. This extra information could allow the model to predict KCNQ1 gating parameters with further improved accuracy than ANNs trained on a single, static structure and address the effect side chain packing on protein stability. Furthermore, it will be valuable to investigate whether the dynamical properties of KCNQ1 protein determined in molecular dynamics simulations, can be used for interpreting the effects of variants.

## Conclusion

We developed a model using biophysical features that can predict the functional consequence of KCNQ1 variants with comparable accuracy to a model that uses using PSSM-based

evolutionary features. We found that combining evolutionary and biophysical features together created optimal model performance. We used biophysical features derived from a three-dimensional structure of KCNQ1 and demonstrated these features can be employed to develop a functional prediction method, highlighting vital structure-function relationships. Moreover, Q1VarPredBio will be a helpful tool to evaluate variants of uncertain significance and improve the accuracy of genetic diagnoses for LQTS. Q1VarPredBio is publicly available as a webserver at www.kcnq1predict.org.

## Method and material

### Neural network architecture and training

A fully connected multitask feed forward ANN with a leaky rectifier linear unit was utilized for all the models. The number of nodes in the input layers was equal to the number of predictive features i.e., 14 for biophysical features, two for evolutionary features, and 12 for biophysical and evolutionary features together. The output layer of each network had four neurons, one for each phenotype prediction for functional parameters of KCNQ1. ANNs were trained with dropouts to prevent overfitting. For instance, the introduction of dropouts in the input layer and hidden layers improve average MCC for the biophysical model from 0.38 to 0.49. The first hidden layer with 32 neurons was found optimal for all three models with a dropout rate of 33%. For the networks trained with evolutionary features and biophysical plus evolutionary features, twelve neurons in the second hidden layer with a dropout of 33% were found optimal whereas for the model trained with biophysical features, eight neurons in the second hidden layer without the dropout rate performed better. Additionally, the biophysical model had a 20% dropout rate in the input layer whereas a 5% dropout rate was sufficient to prevent over-fitting for evolutionary model and evolutionary plus biophysical model. These three networks were trained on binary labels (1 for dysfunctional and 0 for normal) for each phenotype with backpropagation of errors. Based on these errors, weights were updated for 1200 iterations with a learning rate set to 0.001 and momentum equal to 0.5. We utilized accuracy at 0.5 as our objective function during training. All neurons utilized a leaky rectifier transfer function

$$f(x) = \begin{cases} x & x > 0 \\ 0.05 & x \le 0 \end{cases} \tag{1}$$

where x is the total input to a neuron. We observed that introduction of second hidden layer improved the performance of the models on non-perturbing mutations with no effect on the prediction accuracy for functional variants.

For better generalizability and balancing the different classes within the training, monitoring, and independent subsets, we adopted a 25-fold cross-validations strategy, wherein 23 subsets (typically 432 variants) were utilized for training, one subset for monitoring (19 variants), and one subset for prediction (19 variants). For further balancing the different classes and removing any effect of biased subset used for prediction and monitoring, we randomly shuffled our data before dividing them into 25 subsets. It was observed that decreasing the size of these subsets hampers the performance suggesting an incomplete dataset for training the model. Moreover, training data was over-sampled with experimental data (= 125) at a ratio of 3:1 for training the model, restricting the model to overtrain on non-perturbing mutations (= 345).

## Performance metrics

The presence of size-imbalanced classes in the dataset led us to adopt Matthew's Correlation Coefficient (MCC) which is proven as the best performance metric available especially when classes in data are imbalanced [48]. It considers all four parameters of the confusion matrix: numbers of true positives (TP), true negatives (TN), false positives (FP), and false negatives (FN) as shown in Eq (1). MCC value of 1 signifies perfect classification, the value of 0 indicates random classification, and the value of -1 means opposite classification. MCC measures the correlation of predicted value with observed value at a specific threshold. MCC values were computed by using BCL (see S1 Protocol Capture) wherein the threshold was adjusted individually for $I_{Ks}$, $V_{1/2}$, $\tau_{act}$ and $\tau_{deact}$ from 0 to 1 to achieve the best MCC for each phenotype. This hyper-parameter fluctuated roughly 20% based on different instances of the model.

$$MCC = \frac{TP \times TN - FP \times FN}{\sqrt{(TN + FN)(TP + FP)(TP + FN)(TN + FP)}} \tag{2}$$

Moreover, to measure the robustness of our predictive models without being dependent on this threshold value, we utilized a receiver operating characteristic (ROC) curve that summarizes the performance of different feature sets on the positive class. In ROC plots, the x-axis indicates a false positive rate (FPR), and the y-axis indicates a true positive rate (TPR). The area under the curve (AUC) in a ROC plot quantifies the performance of the model which can be utilized to compare different models. Higher the AUC, the better the model is in distinguishing between normal (negative class) and dysfunctional (positive class) phenotypes. AUC value more than 0.5 signify that the classifier is better than a random classifier (AUC = 0.5) in distinguishing dysfunctional variants from normal variants. Similarly, AUC value less than or equal to 0.5 indicates that the classifier is unable to distinguish between positive and negative classes.

$$TPR = \frac{TP}{TP + FN} \tag{3}$$

$$FPR = \frac{FP}{FP + TN} \tag{4}$$

We evaluated the ability of evolutionary features, biophysical, and the combination of both the features to correctly classify the dysfunctional versus normal variants by plotting ROC curves to compare feature sets by their binary classification capability. The ROC curves for 25-fold cross validated models are shown in Fig 3 with shaped region depicting 99% confidence interval.

## Input sensitivity analysis

The prediction of phenotypes for functional parameters are average predictions of 25 independent subsets using 25 monitoring subsets. We can analyze the average effect of input features on these 25 prediction datasets using the concept of input sensitivity. However, we acknowledge from Brown et al. [43] that calculating the magnitude of input sensitivity for feature importance cannot be meaningfully used due to the issue in rescaling the input features. Thus, we recall the consistency method adopted by Brown et al. [43] to evaluate the consistency of feature perturbation on our four result labels across the cross-validation models. Here, we iterate across all input features of the training set and change feature value by a small amount to record the movement of the result label. For each feature with a corresponding label, we count the number of models that will improve the prediction by a change in the descriptor. The net

consistency is defined as zero when half of the variants predict a positive change with respect to the result label and the other half predicts a negative change with respect to the result label. This sensitivity result reported in the result section is averaged across all the 320 models simulated by shuffling dataset 320 times, with each model individually averaged for 25 cross validation models, and instances in the training dataset for each individual feature for desired result label.

## Supporting information

**S1 Fig. Distance from the channel pore axis.**
(DOCX)

**S2 Fig. Depth of the site of mutation on the membrane.**
(DOCX)

**S3 Fig. Definition of three regions of hydrophobicity utilized in this work.**
(DOCX)

**S4 Fig. This figure depicts polarizability distribution with clusters of high, medium, and low polarizability.** This also captures the concept of functional density by quantifying these clusters of polarizabilities for different neighborhood size.
(DOCX)

**S5 Fig. Correlation of peak current with polarizability at different pockets in the protein structure.**
(DOCX)

**S6 Fig. A definition of neighbor that includes a smooth transition function used in the neighbor vector algorithm with lower bound at 3.3 Å and upper lower at 11.4 Å.**
(DOCX)

**S7 Fig. Distribution of Prediction by ANN for non-perturbing, benign, and pathogenic variants depicting that ANN can distinguish these variants by predicting in three different regions between 0 and 1.** Decision threshold is between benign and pathogenic variants.
(DOCX)

**S8 Fig. Percentage of accurate predictions for GOF, LOF and WT-like based on peak current density by the three ANNs models considered in this study.**
(DOCX)

**S9 Fig. Exclusion of 3 biophysical features and 1 evolutionary feature does not affect the performance when evolutionary and biophysical features are combined.**
(DOCX)

**S1 Data. Dataset used for this work.**
(XLSX)

**S1 Protocol Capture. Protocol capture using BioChemical Library(BCL).**
(DOCX)

**S1 Text. Text on extraction of evolutionary and biophysical features.**
(DOCX)

**S1 Table. Amino acid parameters.**
(DOCX)

## Acknowledgments

We acknowledge constructive discussions with Dr. Bian Li.

## Author Contributions

**Conceptualization:** Saksham Phul, Georg Kuenze, Jens Meiler.

**Data curation:** Saksham Phul, Carlos G. Vanoye.

**Funding acquisition:** Charles R. Sanders, Alfred L. George, Jr., Jens Meiler.

**Investigation:** Saksham Phul.

**Methodology:** Saksham Phul, Georg Kuenze.

**Project administration:** Saksham Phul.

**Resources:** Jens Meiler.

**Software:** Saksham Phul.

**Supervision:** Georg Kuenze, Jens Meiler.

**Validation:** Saksham Phul.

**Visualization:** Saksham Phul.

**Writing – original draft:** Saksham Phul.

**Writing – review & editing:** Saksham Phul, Georg Kuenze, Carlos G. Vanoye, Charles R. Sanders, Alfred L. George, Jr., Jens Meiler.

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
