## [Decision Letter · Decision Letter 0]

17 Jan 2022

Dear Dr. Meiler,

Thank you very much for submitting your manuscript "Predicting the Functional Impact of KCNQ1 Variants with Artificial Neural Networks" for consideration at PLOS Computational Biology.

As with all papers reviewed by the journal, your manuscript was reviewed by members of the editorial board and by several independent reviewers. In light of the reviews (below this email), we would like to invite the resubmission of a significantly-revised version that takes into account the reviewers' comments.

We cannot make any decision about publication until we have seen the revised manuscript and your response to the reviewers' comments. Your revised manuscript is also likely to be sent to reviewers for further evaluation.

Sincerely,

Joanna Slusky, Ph.D.

Associate Editor

PLOS Computational Biology

Arne Elofsson

Deputy Editor

PLOS Computational Biology

Reviewer's Responses to Questions

**Comments to the Authors:**

Reviewer #1: The manuscript by Phul et al describes the development of an ANN to predict functional consequences of variants in the KCNQ1 ion channel. It is well written and mostly easy to follow. The results are clearly described and I only have few reservations (see below). Most importantly, I would like to understand a bit better the authors’ negative dataset and how this influences the practical utility of the method.

Major

1.

The authors state in the abstract that the functional predictions “can assist physicians in taking appropriate treatment decision for patients with genetic heart rhythm disorder.” This may well be true, but it’s hard to assess given the information in the paper. Importantly (as the authors are aware), clinical utility requires more than accurate predictions, including that they are actionable. I believe KCNQ1 is, but that is not clear from the paper. In addition, the prediction methods would need to be included in assigning variant effects, and again it is unclear whether the methods are better than what’s out there. Thus, while I don’t doubt that the methods developed are both interesting and could become useful, I would ask that the authors provide a bit more detail how these methods can actually help in treatment or remove this from the abstract. Although only a wording issue, I have added this in “major comments” section as I think it is important.

2.

I am not familiar with the genetics of LQTS/LQT1, so it would be great if the authors could comment on whether all the disease variants are dominant, recessive, work via compound heterozygosity or what the genetics is. In particular, I would like them to provide evidence that training a model based on experimental data of homozygous cells is relevant for the disease (in particular since I assume KCNQ1 is a tetramer).

3.

It would be useful if the authors could explain better both the idea behind their “silent” variants and how they control for that they don’t lead to artificial improvements. I assume the problem is that there are not sufficiently many known benign variants (or variants that have “neutral” phenotypes according to the criteria in Table 1). I couldn’t easily find the numbers, but would be good to know exactly how many “real” “neutral/benign” variants they have. Also, given that many of the features would be zero (I think) for these silent variants, how do the authors know that the ANN isn’t just learning to recognize this? Does the method work as well to recognize real neutral variants as it does for the artificial ones? In this regard, I am curious about the statement:

“We observed that introduction of second hidden layer improved the performance of the models on silent mutations with no effect on the prediction accuracy for functional variants.” Ideally, the authors should provide some assessment of accuracy for actual benign variants even if the method is trained on the artificial ones also.

4.

If I understand correctly, in their predictions, the authors lump together loss of function and gain of function variants as one class. Is that correct? It would have been useful to know how many are in each class, and how the predictions work for them separately? I would e.g. have guess that the evolution-based methods work better for LOF than for GOF.

5.

The authors use a relatively simple PSSM-based method to estimate variant effects from the alignments. That’s of course completely valid, but as they are well aware there are plenty of more advanced methods out there that use more information in the alignments. I would recommend that the authors either use or benchmark with these methods or alternatively tone down the fact that the structure/biophysics-based methods are “better”. Perhaps just say that they are better than using a PSSM, but that more advanced sequence-based methods are known to provide better predictions than PSSMs (at least in many cases).

6.

The authors write “MCC values were computed by using Biochemical Library (BCL) wherein the threshold was adjusted individually for IKs, V1/2, τact and τdeact from 0 to 1 to achieve the best MCC for each phenotype.”. How precisely were these hyper-parameters chosen and did the authors do anything to avoid overfitting?

7.

One of the main points from the authors is that the predictions based on biophysical features can be used to gain mechanistic insight. It would this be useful and interesting with some examples of this. The input-sensitivity analysis provides a bit of this, but it’s not really very detailed. This is complicated further by the fact that many of these scores may be correlated with each other, and given that they use relatively many features could make it difficult to pinpoint specific effects. That said, I think it would be relevant if the authors could make this part a bit clearer and perhaps discuss some examples of variants that they think they can explain why cause LOF.

8.

I thank the authors for making a web-server available. That’s great. I would recommend that they also make their code available so that others in principle can look underneath the hood. Also, it would be good to store the multiple sequence alignment that is used to calculate the PSSM.

Minor:

p. 4, line 70: The authors write “membrane proteins” but refer only to data for KCNQ1 in LQT1. I suggest either making it clear that it’s only a single protein or provide references for other proteins whose destabilization and mistrafficking is involved in LQT1.

p. 4, line 81–82: The authors write “but these tools have limited applicability for KCNQ1”. Could they provide a citation or evidence for this?

p. 5, line 99: I’ll let it be completely up to the authors, but I would recommend deletion of “advanced and innovative”.

p. 13: The authors write “We found that three out of 14 features used for biophysical model failed to improve the predictions on the unseen dataset, therefore were excluded”, but I couldn’t see this analyses.

Reviewer #2: In the present manuscript, the authors used machine learning to develop an advanced method for functional prediction of KCNQ1 mutants. The functional classification categories of this method were then expanded by a scheme that was able to predict variant-specific changes in four electrophysiological KCNQ1 parameters (namely peak current density, voltage of half-maximal activation, and activation and deactivation time constants). The authors then evaluated the performance of artificial neural networks trained on evolutionary and biophysical features for KCNQ1 and observed that a combination of both features produced a model with good predictive accuracy.

This study is complementary to several previous studies in which artificial intelligence was used to predict protein structure and the properties of its mutants.

The concept behind this paper have merit. In fact, this concept is intriguing and has the potential to provide important new information for both basic (mode of action) and applied science (therapy).

Although this manuscript is original and novel, one major issue is that in this study, only a single type of protein is being tested as a case study which questions the generalization of the results. The authors should therefore test an additional type of protein to make the results and conclusions more universal.

In addition, the authors trained and also validated their models with 125 KCNQ1 variants that were functionally evaluated by others; there is thus a need for more functional data – for example, for variants within different domains of the protein. The authors should test/validate their prediction using mutants that are not included in the training sets.

In general, I think that the computational experiments and analysis described are developed sufficiently within the manuscript to inspire confidence that the findings produce meaningful results, and as mentioned above, this paper describes an interesting concept but the fact that this analysis was done for only a single type of protein and that training and validation were performed on the same variants are a major issues and must be addressed before acceptance of this paper for publication.

**Have the authors made all data and (if applicable) computational code underlying the findings in their manuscript fully available?**

Reviewer #1: **No: **

Reviewer #2: None

PLOS authors have the option to publish the peer review history of their article (what does this mean?). If published, this will include your full peer review and any attached files.

Reviewer #1: No

Reviewer #2: No
---

## [Decision Letter · Decision Letter 1]

18 Mar 2022

Dear Mr. Phul,

We are pleased to inform you that your manuscript 'Predicting the Functional Impact of KCNQ1 Variants with Artificial Neural Networks' has been provisionally accepted for publication in PLOS Computational Biology.

Best regards,

Joanna Slusky, Ph.D.

Associate Editor

PLOS Computational Biology

Arne Elofsson

Deputy Editor

PLOS Computational Biology

Reviewer's Responses to Questions

**Comments to the Authors:**

Reviewer #1: I thank the authors for their detailed responses to my comments and the updates to the paper. I already liked the original version and find the revised version strengthened and more clear. Congratulations on a nice piece of work.

Reviewer #2: Accept.

**Have the authors made all data and (if applicable) computational code underlying the findings in their manuscript fully available?**

Reviewer #1: Yes

Reviewer #2: Yes

PLOS authors have the option to publish the peer review history of their article (what does this mean?). If published, this will include your full peer review and any attached files.

Reviewer #1: No

Reviewer #2: No

---

## [Editor Report · Acceptance letter]

14 Apr 2022

PCOMPBIOL-D-21-02206R1 

Predicting the Functional Impact of KCNQ1 Variants with Artificial Neural Networks

Dear Dr Phul,

I am pleased to inform you that your manuscript has been formally accepted for publication in PLOS Computational Biology. Your manuscript is now with our production department and you will be notified of the publication date in due course.

With kind regards,

Anita Estes
